# Genome-Wide Analysis of the WRKY Transcription Factor Family in Roses and Their Putative Role in Defence Signalling in the Rose–Blackspot Interaction

**DOI:** 10.3390/plants13081066

**Published:** 2024-04-10

**Authors:** Helena Sophia Domes, Thomas Debener

**Affiliations:** 1Department of Molecular Plant Breeding, Institute for Plant Genetics, Leibniz Universität Hannover, 30419 Hannover, Germany; 2Julius Kühn-Institut, Federal Research Centre for Cultivated Plants, Institute for National and International Plant Health, 38104 Braunschweig, Germany

**Keywords:** *Diplocarpon rosae*, *Rosa chinensis*, RNA-Seq, biotic stress, fungal infection, WRKY

## Abstract

WRKY transcription factors are important players in plant regulatory networks, where they control and integrate various physiological processes and responses to biotic and abiotic stresses. Here, we analysed six rose genomes of 5 different species (*Rosa chinensis*, *R. multiflora*, *R. roxburghii*, *R. sterilis*, and *R. rugosa*) and extracted a set of 68 putative *WRKY* genes, extending a previously published set of 58 *WRKY* sequences based on the *R. chinensis* genome. Analysis of the promoter regions revealed numerous motifs related to induction by abiotic and, in some cases, biotic stressors. Transcriptomic data from leaves of two rose genotypes inoculated with the hemibiotrophic rose black spot fungus *Diplocarpon rosae* revealed the upregulation of 18 and downregulation of 9 of these *WRKY* genes after contact with the fungus. Notably, the resistant genotype exhibited the regulation of 25 of these genes (16 upregulated and 9 downregulated), while the susceptible genotype exhibited the regulation of 20 genes (15 upregulated and 5 downregulated). A detailed RT–qPCR analysis of *RcWRKY37*, an orthologue of *AtWRKY75* and *FaWRKY1*, revealed induction patterns similar to those of the pathogenesis-related (PR) genes induced in salicylic acid (SA)-dependent defence pathways in black spot inoculation experiments. However, the overexpression of *RcWRKY37* in rose petals did not induce the expression of any of the PR genes upon contact with black spot. However, wounding significantly induced the expression of *RcWRKY37*, while heat, cold, or drought did not have a significant effect. This study provides the first evidence for the role of *RcWRKY37* in rose signalling cascades and highlights the differences between *RcWRKY37* and *AtWRKY75*. These results improve our understanding of the regulatory function of WRKY transcription factors in plant responses to stress factors. Additionally, they provide foundational data for further studies.

## 1. Introduction

The WRKY transcription factor (TF) family is one of the largest TF families in plants and plays important roles in a variety of physiological processes. WRKY family members are involved in the signalling cascade of responses to external biotic or abiotic factors, as well as the regulation of internal plant processes such as growth, senescence, and seed development. WRKY TFs act as both activators and repressors [1].

The members of the WRKY TF family have a WRKY domain consisting of approximately 60 amino acids, with the conserved sequence WRKYGQK and a zinc finger-like motif [2]. Based on these structures, the different WKYY TFs can also be divided into three groups. Group I proteins have two WRKY domains instead of one as in the other groups. Group 2 is further subdivided into subgroups (a–e) [3]. Group 3 proteins differ in their zinc finger motifs (C2HC instead of C2H2).

The WRKY domain enables these TFs to bind to W-box elements (TTGACC/T) in promoter regions of target genes [3,4]. WRKYs play an important role in the signalling during a pathogen attack. For example, pathogenesis-related (PR) genes, which play a central role in the activation of defence mechanisms, have W-box elements in their promoters so that their expression can be influenced by WRKYs (e.g., [5,6,7]). Much is known about the function of WRKY TFs in the immune response in *Arabidopsis*, rice, and tobacco.

More than half of all *Arabidopsis WRKYs* respond to infection or endogenous salicylic acid (SA) application [8]. A link to the SA-dependent immune response is also supported by Wang et al. [9], who found that *AtNPR1* directly upregulates eight *WRKYs*. Direct and indirect regulation by mitogen-activated protein kinases (MAPKs) has also been reported [10,11,12]. For example, *AtWRKY33*, a substrate of MPK3/MPK6, is essential for the downstream production of the phytoalexin camalexin in response to pathogens [13].

However, little is known about the function of WRKY TFs in the family *Rosaceae*. A more detailed review of WRKY Group I candidates was provided by Jiang et al. [14] for seven *Rosaceae* species (*Fragaria vesca*, *Malus domestica*, *Prunus persica*, *Pyrus bretschneideri*, *Prunus mume*, and *Rubus occidentalis*), which can serve as a basis for further studies. For *Prunus mume* [15] and strawberry [16], more detailed analyses of the entire WRKY family were also carried out. Based on the genome provided by Raymond et al. [17], Liu et al. [18] described 56 candidate *WRKYs* of *Rosa chinensis* in more detail and analysed their expression after *Botrytis cinerea* infection. In particular, they found that the homologue of *AtWRKY33* is involved in the response to fungal infection.

In *Fragaria × ananassa*, a member of the family *Rosaceae*, *FaWRKY1*, the homologue of *AtWRKY75*, has been described as an important component of signalling following infection with the fungus *Colletotrichum acutatum* [19]. Following inoculation with powdery mildew (*Podosphaera aphanis*), Wei et al. [20] observed regulatory changes in 45 of 62 *Fragaria vesca WRKYs*. Approximately 2/3 of these genes were upregulated.

Consistent with this, Chen et al. [21] considered *AtWKY75* together with *AtWRKY28* to be positive regulators of SA- and jasmonic acid (JA)/ethylene (ET)-dependent defence signalling networks, with a focus on the JA/ET pathway in response to the necrotrophic fungus *Sclerotina sclerotiorum*.

In the present work, we compiled a list of *WRKY* candidate genes in the genus *Rosa* by extending the set analysed by Liu et al. [18] by searching five other genomes (*R. chinensis* [22], *R. rugosa* [23], *R. multiflora* [24], *R*. *roxburghii*, and *R. sterilis* [25]). In addition, the expression of these candidate *WRKYs* following fungal infection was determined by RNA-Seq data from infected and mock-inoculated rose leaves. Furthermore, the expression of *RcWRKY37*, the homologue of *AtWRKY75* and *FaWRKY1*, was determined in further inoculation experiments, overexpression studies, and after the application of abiotic stress factors. In these inoculation experiments, we also closely examined the expression of PR genes and marker genes from other defence-related signalling pathways.

The hemibiotrophic fungus *Diplocarpon rosae*, which causes rose black spot disease, was used as the pathogen. This disease has a major impact on rose cultivation and presents as chlorotic or necrotic spots on leaves, spreading in a characteristic star pattern [26]. Infestation with *D. rosae* can result in early defoliation and reduce the ornamental value of affected plants [27]. Under favourable conditions, fungal spores can germinate on plant leaves within eight hours. They then penetrate the leaf cuticle, forming haustoria within plant cells. Approximately six days after infection, necrotrophic intracellular hyphae develop. This cycle continues with the release of new conidia from acervuli, the fungal reproductive structures [28,29,30]. If the interaction is incompatible, fungal development ceases before haustorium formation is complete [31].

Our investigation is particularly important for elucidating the defence mechanisms and providing potential avenues for crop improvement, given the economic importance of roses and the substantial damage inflicted by *D. rosae* annually.

## 2. Materials & Methods

### 2.1. Plant and Fungal Material

In this study, plants of the rose cultivar ‘Pariser Charme’ (PC), a line of this cultivar exhibiting the stable integration of the *D. rosae* resistance gene *Rdr1* (PC::*Rdr1*, [31]), and the resistant rose genotype 91/100-5 [32] were used. PC is a hybrid tea variety (*Rosa* ×*hybrida*), of which *R. chinensis* is one of the progenitors, and genotype 91/100-5 originates from a cross between a hybrid tea variety and an *R. multiflora* hybrid. The fungal pathogen *D. rosae* Wolf (isolate DortE4; [33]) was propagated on susceptible PC leaves for use in this study [27].

### 2.2. Inoculation Assay and Experiments

Freshly unfolded rose leaves were detached and spray inoculated with a 500,000 *D. rosae* conidia/mL suspension or water. Leaves were stored in translucent boxes on moist tissue at 20 °C. For further information on the inoculation assay and sample collection as well as photographs, see the description in Neu et al. [34].

A description of the 3′ mRNA-Seq dataset used, including the inoculated PC and 91/100-5 genotypes, can also be found in Neu et al. [34]. The MACE (Massive Analysis of cDNA Ends; [35]) technique was used to generate transcriptomic data. All of the raw data are available from the NCBI under BioProject number PRJNA445241. The heatmap was generated using the R package ‘pheatmap’ [36].

Another inoculation experiment with the resistant PC line PC::*Rdr1* and normal PC was evaluated using the BioMark high-throughput PCR/qPCR system (Fluidigm Corporation, San Francisco, CA, USA). Leaf samples were collected at 0, 12, 48, and 72 h post infection (hpi) and frozen in liquid nitrogen. The experiment was performed in three time-independent replicates.

### 2.3. Identification of WRKY Candidates in R. chinensis and Other Rose Species

The genomes and gene predictions of *R. chinensis* [17,22], *R. multiflora* [24], and *R. rugosa* [23] can be found at www.rosaceae.org (accessed on 24 November 2021). The genomes of *R. roxburghii* and *R. sterilis* can be found at https://db.cngb.org/cnsa/ ([25], accessed on 20 December 2023). The sequences (nucleotide and protein) of the *Arabidopsis* WRKYs (www.arabidopsis.org; accessed on 24 November 2021) and the conserved WRKY domain were used as a basis to search for WRKY candidates. BLAST searches were performed locally using BioEdit ([37], version 7.2.5). The chromosome map was generated using MagGene2Chrom ([38], version 2.1). All candidates were checked for the presence of at least one WRKY domain [39]. The protein sequences of the roses used can be found in the Appendix A.

### 2.4. Phylogenetic Analyses

MEGA X was used to conduct phylogenetic analysis [40]. The required alignments were performed using the MUSCLE alignment software [41] version implemented in MEGA X with default settings. Phylogenetic trees were produced from the protein alignments using the maximum likelihood method and the JTT matrix-based model [42] with 1000 bootstrap replicates. A graphical representation was generated using iTOL ([43], version 6.9). A phylogenetic analysis of all used WRKY protein sequences was performed as described above and can be found in the Appendix A.

### 2.5. Promoter Region Analysis

For the analysis of the promoter regions of *R. chinensis WRKYs*, sequences 1500 bp upstream of the respective gene start were excised. For genes located on the minus strand, the region was examined as a reverse complement. The analysis was performed using plantCARE ([44], accessed on 2 December 2021) on both the plus and minus strands. The graphical representation was generated using the Gene Structure Display Server (http://gsds.gao-lab.org/, accessed on 13 December 2021) of Hu et al. [45].

### 2.6. Overexpression Experiments

To test the role of *RcWRKY37* in gene regulation, a transient overexpression experiment was performed. The complete gene sequence of *RcWRKY37* was transformed into a binary vector (C757pGFPU10-ocs-LH, DNA cloning service Hamburg, Germany) by In-Fusion^®^ cloning (Takara Bio Inc., Ltd., Shiga, Japan) and sequenced as a control. The transgene was overexpressed through the upstream 35S promoter. GFP under the control of the *A. thaliana* ubiquitin promoter was used as a marker gene. Transient overexpression was induced by infiltration of PC petals with *Agrobacterium tumefaciens* (strain GV3101), as previously described [46]. Gene expression was always determined relative to petals infiltrated with empty vector *A. tumefaciens*. Samples were collected 4 days after infiltration when a good GFP signal was visible. The experiment was performed in three independent replicates with three biological replicates each.

The leaf tissue frozen in liquid nitrogen was disrupted using a QIAGEN TissueLyser II (Venlo, The Netherlands). RNA purification was carried out using the RNeasy^®^ Plant Mini Kit from Qiagen (Hilden, Germany), with additional DNase digestion provided by an Ambion VR DNA-free™ Kit from Life Technologies (Carlsbad, CA, USA) to eliminate residual genomic DNA, following the manufacturer’s protocol. For cDNA (LunaScript^®^ RT SuperMix Kit, New England Biolabs, Ipswich, MA, USA), 300 ng of RNA was used and diluted with ddH_2_O at 1:5 before qPCR. The qPCR analyses and data processing were carried out using the QuantStudio™ 6 Flex System (Applied Biosystems, Austin, TX, USA) and the Luna^®^ Universal qPCR Master Mix (New England Biolabs, Ipswich, MA, USA). The annealing temperature was set at 64 °C. PCR primers were designed for gene prediction in the *R. chinensis* genome [22] using primer3plus [47], and the results are provided in Appendix A, along with their efficiencies.

### 2.7. Experimental Induction of Abiotic Stress

Young and freshly unfolded leaves of the variety PC were placed onto moist tissues in transparent containers and subjected to different types of stress: cold (5 °C), heat (37 °C), wounding (punctured several times with a scalpel), and a control (21 °C). The “drought” treatment was performed in open boxes without moisture. The stressors were applied for one hour in darkness, followed by freezing the samples with liquid nitrogen for subsequent analyses. Three biological replicates were analysed per treatment in three independent experiments.

RNA isolation, cDNA synthesis, and qPCR analysis were carried out as described in Section 2.6, except that RNA was isolated using the Quick-RNA™ MiniPrep Plus kit (Zymo Research, Irvine, CA, USA), according to the manufacturer’s instructions with minor modifications. The lysis buffer was supplemented with dithiothreitol (DTT) to a final concentration of 50 mM.

## 3. Results

### 3.1. Identification of an Extended Set of WRKY Candidates by Analysing Six Rose Genomes

We first searched the fully annotated genome of *R. chinensis* provided by Hibrand Saint-Oyant et al. [22] for the identification of candidate *WRKY* genes. To achieve this goal, a BLAST search was carried out with the use of known *Arabidopsis* WRKY sequences and the conserved WRKY domain as references. Through this approach, 112 sequences were successfully identified in the *R. chinensis* genome [22]. All identified sequences were analysed at the protein level to confirm the presence of one or more WRKY domains with the aid of the NCBI CCD database [39]. As a result of this analysis, a set of 68 refined WRKY candidates (Appendix A) containing either smart00774 or pfam03106 domains, both of which belong to the WRKY superfamily Cl03892, was obtained. In the gene predictions for *R. multiflora* and *R. rugosa*, 60 WRKY candidates were found in each case. In *R. roxburghii*, the analyses yielded 56 candidates, and in *R. sterilis*, 67 candidates were identified. This extended the published set of WRKY candidates in roses by 10 additional sequences.

*WRKY* genes are located on all chromosomes of *R. chinensis* (Figure 1). Nonetheless, the assignment of two *WRKY* gene models (RC0G0015100 and RC0G0183100) to particular chromosomes in the genome of Hibrand Saint-Oyant et al. [22] was inconclusive (“chromosome 0”). Consequently, the positions of these two sequences were assessed using the genome provided by Raymond et al. [17], leading to their recognition on chromosome 2 as *RcWRKY19* and *RcWRKY20*. The *WRKY* genes were subsequently labelled *RcWRKY1* through *RcWRKY68* based on their position on the seven chromosomes. Chromosomes 1 and 2 each contained 12 *WRKY* genes, while chromosome 3 had 10, chromosome 4 had 6, chromosome 5 had 11, chromosome 6 had 6, and chromosome 7 had 11.

### 3.2. Phylogenetic Analyses

An initial phylogenetic tree was generated with the 68 RcWRKY sequences from Appendix A and the 73 identified AtWRKY proteins, including their isoforms (Figure 2).

### 3.3. Expression of the RcWRKY Genes under Pathogen Attack

To investigate the expression patterns of the identified rose *WRKY* candidates under biotic stress conditions, we analysed data from a previous RNA-Seq experiment [34]. Leaves of the susceptible rose cultivar PC were inoculated with the fungus *D. rosae* in a detached leaf assay. Notably, a previous publication by Neu et al. [34] provided a general evaluation of the expression data derived from this experiment. In addition to the analyses presented in the aforementioned publication, we further investigated the expression profiles of samples from the rose genotype 91/100-5, which exhibits resistance against *D. rosae*. The heatmap presented in Figure 3 provides an overview of the regulatory changes observed in the *RcWRKY* genes relative to those in mock-inoculated leaves.

Following inoculation with *D. rosae*, regulatory changes were evident in *RcWRKYs* in both susceptible and resistant genotypes. Among the analysed *WRKYs*, a total of 9 genes were downregulated, while 18 were upregulated in response to the pathogen. Notably, the expression of 41 *RcWRKYs* did not significantly change upon pathogen infection. Many of the regulated genes are affected by transcriptional changes in both genotypes. The exceptions were *RcWRKY7* (RC1G0496700), *RcWRKY31* (RC3G0309600), *RcWRKY51* (RC5G0738400) and *RcWRKY68* (RC7G0571000) for the downregulated genes and *RcWRKY23* (RC2G0629000) and *RcWRKY42* (RC5G0107600) for the upregulated genes, which are all only regulated in the resistant 91/100-5 genotype. *RcWRKY57* (RC6G0593600) is the only gene that only showed a regulatory change in PC.

The *AtWRKY75* homologue Rc*WRKY37* (RC4G0344000) is a candidate that is upregulated after pathogen attack but also differs between susceptible and resistant genotypes. Although both genotypes exhibited upregulation of *RcWRKY37* expression 24 h after *D. rosae* inoculation, the gene was no longer significantly upregulated in the incompatible interaction after 72 h.

### 3.4. Promoter Analyses

To gain insight into the potential signalling pathways in which the identified RcWRKYs may be involved, a comprehensive analysis of the promoter regions of all 68 sequences was performed. The promoter regions were analysed with plant CARE, considering both the plus and minus strands. The focus of the analysis was on identifying motifs associated with stress response and phytohormone signalling. For abiotic stresses, motifs related to low temperature (LTR) and drought responsiveness (MBS) were selected and investigated. In terms of defence response, *cis*-acting elements such as TC-rich repeats and the distinctive WRKY TF-binding site, known as the W-box, were analysed. Several motifs involved in phytohormone signalling have been explored. For abscisic acid (ABRE), auxin (AuxRR-core, TGA-box, TGA-element), gibberellin (GARE-motif, P-box, TATC-box), JA (CGTCA-motif, TGACG-motif) and SA (SARE, TCA-element) were investigated. The motifs discovered through this analysis are presented separately for the plus and minus strands in Figure 4 and Figure 5, respectively.

In the analysed promoter regions, stress-related *cis*-elements were identified in all 68 sequences. The most prevalent motif observed was ABRE, indicating a connection to abscisic acid (ABA) signalling. ABREs were found in 79.4% of the promoter sequences, with 58.8% on the plus strand and 51.5% on the minus strand. In several cases, multiple ABRE motifs were present within a single promoter sequence. Notably, motifs associated with JA responsiveness were also highly prevalent, occurring in 69.1% of the sequences overall (67.6% on both the plus and minus strands).

Similarly, motifs linked to the biotic stress response were observed with a comparable frequency (66.2% overall, 47.1% on the plus strand, and 51.5% on the minus strand). Motifs associated with the gibberellin response were present in just over half of the sequences (52.9%, with 33.8% on the plus strand and 27.9% on the minus strand). The remaining three groups—auxin-related motifs (39.7% overall, 16.2% on the plus strand, and 26.5% on the minus strand), drought-responsive motifs (38.2% overall, 20.6% on the plus strand, and 25% on the minus strand), and low-temperature-responsive motifs (35.3% overall, 23.5% on the plus strand, and 14.7% on the minus strand)—were found with lower frequencies.

In the promoter region of the *RcWRKY37* gene (RC4G0344000), the *AtWRKY75* homologue, three ABREs (ABA-related) were found on both the plus and minus strands. Moreover, on the minus strand, two additional TGA elements and a W-box motif, and on the plus strand, one CGTCA motif (JA-related) was also discovered.

Table 1 presents a comprehensive analysis of the promoter regions of differentially expressed and nonregulated *WRKY*s following infection with *D. rosae*, as depicted in Figure 3. In the table, the distinction between plus and minus strands has been omitted to present a consistent view of the results.

ABREs were more common among the regulated genes. Specifically, this motif is present in only 73.2% of the promoter regions of nonregulated *WRKYs*. In contrast, 83.3% of the *WRKYs* were upregulated, and 100% of the *WRKYs* were downregulated.

Among the auxin-related motifs, there was a notable difference in the abundance of TGA elements among the differentially expressed *WRKYs*. Specifically TGA elements were more prevalent in the promoter regions of downregulated *WRKYs*, with a frequency of 55.6%. In comparison, the frequency of TGA elements was approximately the same for upregulated genes (27.8%) and nonregulated genes (26.8%).

The TATC-box motif, which is associated with the gibberellin response, exhibited a slightly different pattern. The presence of this motif in the promoter region of *WRKY* genes varies depending on their regulation status. In particular, the proportion of *WRKYs* with the TATC-box motif was greater in the promoter regions of upregulated genes (16.6%) and nonregulated genes (14.6%), while none of the downregulated *WRKYs* contained this motif (0%). For the MBS motif, there was an increased frequency of upregulated genes (50%) compared to downregulated (33.3%) and nonregulated (34.1%) genes.

Distinct differences were also observed among the biotic stress-related motifs. This disparity is particularly evident for the W-box motif, which was present in the promoter regions of 72.2% of the upregulated *WRKYs*. In *WRKYs* that were not regulated during *D. rosae* infection, 48.8% of the genes exhibited W-box motifs, while in downregulated genes, this motif was found in only 33.3% of the promoter regions. Furthermore, TC-rich repeats exhibited a slightly greater frequency in the promoter regions of downregulated *WRKYs* (44.4%). For the nonregulated genes, 34.1%, and for upregulated genes, 27.8%.

No significant differences were detected in the distribution of JA-related motifs among the differentially expressed *WRKY* genes. Similarly, for the SA-related motifs, there were no notable variations based on gene expression patterns. However, it is worth mentioning that the presence of the SARE motif, which is associated with the response to SA, was identified in only one *WRKY* gene, *RcWRKY25* (RC3G0073600), was upregulated in response to *D. rosae* infection.

### 3.5. Expression of RcWRKY37 and Other Defence-Related Genes in Compatible and Incompatible Interactions with D. rosae

Because of its regulation under fungal attack and the role of orthologues in previously published plant–pathogen interactions, the *AtWRKY75* homologue *RcWRKY37* (RC4G0344000) was further examined for its changes in expression during an independent inoculation experiment along with the expression of selected PR genes. The experimental setup was similar to that of the RNA-Seq experiment, but resistant transgenic PC plants containing the resistance gene *Rdr1* were used as the resistant genotype (PC::*Rdr1*; 33). Gene expression was determined by high-throughput qPCR. In addition to *RcWRKY37*, various defence-related genes were analysed. The expression changes in PC and PC::*Rdr1* inoculated with *D. rosae* compared to those in uninfected control leaves at all time points are shown in Figure 6.

The data showed that inoculation resulted in strong upregulation of several PR genes in both genotypes. However, the exact expression patterns of *RcPR1*, *RcPR4*, and *RcPR6* were different. In the case of *RcPR1* and *RcPR4*, the pattern was repeated, with peak expression at 24 hpi in the susceptible PC leaves but not until 48 hpi in the resistant PC::*Rdr1* leaves. For *RcPR6*, the peak observed at 48 hpi was also evident in the trend observed for PC::*Rdr1*. However, few significant regulatory changes were detected in the PC leaves.

The regulation of *RcWRKY37* also showed this pattern. The significant upregulation at 24 hpi in the samples of the susceptible genotype (PC) and at 48 hpi for the resistant genotype (PC::*Rdr1*) compared to 0 hpi was detected. The differences between the two genotypes were also significant at these two time points. Lipase-like *PAD4* and enhanced disease susceptibility 1 (*EDS1*) were also significantly upregulated after *D. rosae* infection. However, there were no significant differences between the genotypes. The expression of two ethylene-responsive transcription factors (*ERF113*/*ERF114*), on the other hand, did not significantly change compared to that at 0 hpi, but the expression of the genotypes significantly differed at 48 hpi (*ERF113a*).

### 3.6. Effect of Transient Overexpression of RcWRKY37 on Defence-Related Genes

To further analyse the effects of RcWRKY37, we cloned the coding region into a binary vector under the control of the 35S promoter and conducted transient expression experiments by infiltrating recombinant agrobacteria into rose petals. In three independent experiments, the overexpression of *RcWRKY37* in petals of the susceptible genotype PC was analysed, and the expression of different marker genes was determined (4 dpi). The expression of *RcWRKY37* was 30-fold greater in the treatment group than in the control group. By selecting genes known to be markers of the ET, JA, and SA signalling pathways, the effect of high *RcWRKY37* expression on defence and phytohormone-induced immune responses was investigated (Figure 7). However, none of these genes showed a response to the increased expression level of *RcWRKY37*.

### 3.7. Abiotic Stress Induction of RcWRKY37

To analyse other potential inducing factors in our detached leaf assay, PC leaves were exposed to different abiotic stresses and the expression of certain genes was determined using RT–qPCR. The results are shown in Figure 8.

*RcWRKY37* (RC4G0344000) expression was significantly upregulated after wounding (4.23×, *p* = 0.006). Changes in the applied temperature and drought stress did not lead to any changes in the expression levels of these genes. The expression of *RcPR1* (RC6G0055800) and *RcPR4* (RC3G0391100), which are markers of pathogen infection, did not change. Moreover, *RcPR1* expression was downregulated after high-temperature treatment, but the other changes in the expression of *RcPR1* were not significant. Overall, abiotic stresses resulted in only minor changes in gene expression.

## 4. Discussion

### 4.1. Identification of WRKY Genes in R. chinensis

In a previous study, Liu et al. [18] identified 58 *WRKY* genes from the *R. chinensis* genome published by Raymond et al. [17]. Here, we significantly increased the number of rose *WRKY* genes by combining information from various rose genomes and provided additional information about WRKY biology in roses via bioinformatic and transcriptomic analyses. By first searching the *R. chinensis* genome of Hibrand Saint-Oyant et al. [22] and combining the data with the *R. chinensis* genome of Raymond et al. [17], we identified an additional 10 *WRKY* candidate genes. Two sequences (RC0G0015100 and RC0G0183100) could only be precisely mapped in the genome of Raymond et al. [17]. The difference between the two genomes, which are based on the same diploid, heterozygous *R. chinensis* genotype ‘Old Blush’, is that the DNA used for each of these genome sequences was obtained independently from haploid tissue and may therefore differ in allelic variants. Further differences arose from the different sequencing and assembly strategies of the two projects [17,22]. A comparative list of the sequences from both projects can be found in the Appendix A. Differences in the annotation of the *Rdr1*-TNL gene family in these two genomes have also been found previously [48], indicating that regional differences in the quality of the assemblies might also exist.

Analysis of four other rose species revealed that the number of candidate *WRKYs* differed among the species. Whether this is due to a true difference in the size of the WRKY family in these species or due to differences in the quality of the genome assembly and annotation is difficult to determine. However, we detected only one additional sequence in *R. sterilis* beyond our 68 *WRKY* candidates, suggesting that the maximum number of *WRKY* genes in roses has been reached. This is a note of caution about the gaps in the quality of most plant genomes published to date and illustrates the usefulness of multiple genomes for the analysis of complex gene families.

### 4.2. Transcriptional Regulation of WRKY Genes in Rose Leaves after Infection with D. rosae

Analysis of the gene expression of the susceptible rose genotype PC after infection with *D. rosae* [34] revealed changes in the expression of 20 *RcWRKYs* (5 downregulated and 15 upregulated). For many of these, changes occurred at both 24 and 72 hpi, while for some, they occurred only at one of the time points. In Liu et al.’s studies [18,49], petals of a resistant *R. hybrida* were infected with *B. cinerea*, and RNA-Seq was used to detect expression changes at 30 and 48 hpi. Although 19 *WRKYs* with altered expression were detected, these *WRKYs* were mostly other members of the family. Of the regulated *WRKYs*, six were similarly regulated in the *R. chinensis* and *R. hybrida* datasets, including *RcWRKY37* (*AtWRKY75*, Liu et al.: *WRKY30*), *RcWRKY11*, *RcWRKY24*, *RcWRKY36*, *RcWRKY40*, and *RcWRKY45*. However, unlike Liu et al. [18], nine of their upregulated *WRKYs* were not upregulated in *R. chinensis*, and four were downregulated instead. Among the other 10 *WRKYs*, a change in regulation was found only in inoculated *R. chinensis* leaves. Notably, the candidate regulator gene for grey mould resistance identified by Liu et al. ([18]; RC5G0618300; Liu et al.: *WRKY41*) was not activated in PC leaves after *D. rosae* infection.

The differences in *WRKY* response may be due to the different pathogen lifestyles, with *B. cinerea* adopting a necrotrophic lifestyle and *D. rosae* adopting a hemibiotrophic lifestyle with a biotrophic interaction at the early stages of the interaction. Accordingly, Liu et al. [49] reported that genes related to the brassinosteroid pathway were highly regulated during *Botrytis* infection, while many SA pathway genes were upregulated in susceptible PC leaves during *Diplocarpon* infection at early time points [34]. In addition, the different tissues analysed (petals vs. leaves) might exhibit different responses to pathogens.

In silico analysis of the promoter regions revealed the presence of numerous stress-related *cis*-elements within the 68 *RcWRKYs*. This further suggested that the WRKY TF family may be involved in different stress regulation processes, which is consistent with the findings of other studies. The ABRE motif is widespread in the promoter regions of *WRKYs* in general, and even more so in those of upregulated and especially downregulated *WRKYs* after *D. rosae* inoculation. The fact that many rose *WRKY* promoters have ABRE motifs supports the results of other studies showing that AtWRKY TFs are important in ABA-dependent signalling networks (e.g., [50,51,52]). Adie et al. [53] reported that ABA-related motifs are overrepresented in the promoters of genes that respond to fungal infection. Notably, the ABRE motif frequently appears multiple times within a single promoter sequence. Previous studies have demonstrated that the presence of at least two ABRE motifs in a promoter region is necessary for ABA-mediated induction [54,55,56]. The rose genome examined here fulfils this theoretical requirement, and the high number of motifs strongly suggests that rose *WRKYs* respond to ABA. The increased frequency of motifs in the promoter regions of the regulated *WRKYs* further supports the role of ABA in the response of roses to fungal attack.

W-boxes were also found more frequently in the promoters of the upregulated *WRKYs*. The percentage of sequences with this motif was 72.2% for upregulated and 33.3% for downregulated *WRKYs*. For nonregulated *WRKYs*, a W-box was found in 48.8% of the promoter regions. The W-box motif serves as a binding site for WRKY TFs [57] and is often associated with the response to pathogens (e.g., [4,58]). In addition to the regulation of other *WRKYs*, the regulation of their own gene expression has also been observed [13,59]. This type of auto- and cross-regulation appears to occur in roses and to play a specific role in the upregulation of *WRKYs* following pathogen contact.

This in silico analysis of the rose WRKY family promoters, combined with the expression data from an inoculation experiment, suggests a possible link to the ABA-dependent signalling pathway during *D. rosae* infection and the cross- and autoregulation of WRKYs. In summary, there is strong evidence supporting the involvement of rose WRKYs in the regulation of multiple signalling pathways.

### 4.3. Expression of RcWRKY37 and Other Defence-Related Genes in Susceptible and D. rosae-Resistant Rose Genotypes

A repeat RT–qPCR experiment for *RcWRKY37* in inoculated susceptible and resistant leaves confirmed the RNA-Seq data (Figure 6), which were based on three completely independent inoculation experiments as biological replicates. In this study, transgenic PC plants harbouring the *Rdr1* resistance gene were used as resistant plants (PC::*Rdr1*), and nontransgenic PC plants were used as susceptible plants in further inoculation experiments. This approach led to a reduction in genotype-related differences and reduced false-positive differences between the two treatments.

The strong regulation of *RcPR1*, *RcPR4*, and *RcPR6*, together with the upregulation of *RcEDS1* and *RcPAD4,* in PC indicated an SA-dependent response of roses to black spot disease. This finding was also in line with the RNA-Seq analysis presented by Neu et al. [34]. Furthermore, the expression pattern of *RcWRKY37*, the orthologue of *AtWRKY75*, was more similar to the expression profiles of *PR1*, *PR4*, and *EDS1* but not from that of JA-related genes. However, no SA-related motif was found in the promoter region of *RcWRKY37*. In *Arabidopsis*, upregulation of *AtWRKY75* is considered to be beneficial for the resistance to fungal pathogens [21,60]. Both publications propose that *AtWRKY75* is a component of the JA/ET pathway, but in both cases, only infections involving necrotrophic fungi were examined.

Two studies have investigated the role of the strawberry AtWRKY75 homologue FaWRKY1 in infection with the hemibiotrophic fungus *Colletotrichum acutatum* [19,61]. Encinas-Villarejo et al. [19] postulated that *FaWRKY1* expression had a positive effect on resistance, but this was mainly based on experiments in *Arabidopsis* plants inoculated with *P. syringae*. In *C. acutatum*-infected strawberries, *FaWRKY1* was upregulated, which was also dependent on the infection intensity. Upregulation after SA application was also observed, but the intensity was cultivar- and tissue-dependent. In contrast, Higuera et al. [61] directly examined the function of FaWRKY1 in strawberries and concluded that transient silencing of the gene resulted in increased resistance to *C. acutatum*, while transient overexpression did not further increase susceptibility, confirming the role of FaWRKY1 in this interaction.

### 4.4. The Role of RcWRKY37 in Different Signalling Pathways

During black spot infection, several *WRKY* genes, including *RcWRKY37*, were regulated. This gene was identified as particularly interesting due previous reports linking it to plant–pathogen interactions and due to its reduced expression after the onset of the defence reaction in our RNA-Seq analyses of incompatible interactions. However, the transient overexpression of *RcWRKY37* in the petals of the susceptible rose variety PC did not affect any tested marker genes of the signalling pathways involved in the plant immune response, including ET- and JA-related genes such as *EIN3*, *ERF1*, *MYC2*, and *PDF1-2*, or SA-related genes such as *EDS1*, *PAD4*, *PR1*, *PR4*, *PR6*, and *SAG101*. The influence of *AtWRKY75* overexpression on SA signalling in *Arabidopsis*, including increased *AtPR1* expression, as reported by Guo [62], and the upregulation of *AtPDF1*, as reported by Chen et al. [60], could not be confirmed. This difference in results may be due to variations in defence signalling between *Arabidopsis* and roses or the use of stably transformed mutants instead of transient expression in petals in which Agrobacteria infiltrated but did provoke defence responses themselves. However, our results aligned with those of Higuera et al. [61], who also found no regulatory changes in genes regulated by *C. acutatum* infection (*FaCHI4-2*, *FaCAT*, *FaJAZ4*, *FaJAZ5*, *FaJAZ9*, *FaWHY1*, and *FaWHY2*) when *FaWRKY1* was overexpressed.

The overexpression data suggested that *RcWRKY37* is not the initiator of a phytohormone-dependent immune response. However, these effects might be induced by stressors acting during the infection process. Some evidence for this is based on changes in gene expression after wounding. *RcWRKY37* was upregulated in response to this abiotic stress, whereas no changes were detected for *RcPR1.* This may indicate less specific regulation of *RcWRKY37*, whereas *RcPR1* expression appears to be specifically regulated by the presence of a pathogen. However, both the RNA-Seq and RT–qPCR results revealed that *RcWRKY37* was upregulated following fungal infection in resistant rose plants. The upregulation of *RcWRKY37*, particularly at a later time point compared to the expression in the susceptible genotypes, cannot be attributed to injury by invading fungal hyphae, as these hyphae did not develop further in resistant plants at this stage. *RcWRKY37* appears to be involved in the response to abiotic wounding, as well as in parallel induction upon pathogen infection, but does not seem to be involved in the regulation of SA-related PR genes. An investigation of knockout mutants could provide a more precise understanding of the extent to which RcWRKY37 influences resistance and its location in the signalling chain in roses.

## 5. Conclusions

In conclusion, this study provides valuable insight into the WRKY TF family in roses and its regulation during the early stages of rose black spot interaction. The examination of promoter regions and gene expression patterns indicated the involvement of WRKYs in multiple signalling pathways, including ABA-dependent and SA-dependent pathways. These findings improve our understanding of the molecular mechanisms governing the response of rose plants to fungal pathogens. These findings suggest that *RcWRKY37* does not induce phytohormone-related immune responses beyond the levels caused by Agrobacterium infiltration, as demonstrated by its transient overexpression in PC petals. This overexpression had no effect on marker genes in these pathways. However, *RcWRKY37* was upregulated in response to abiotic wounding.

## Figures and Tables

**Figure 1 plants-13-01066-f001:**
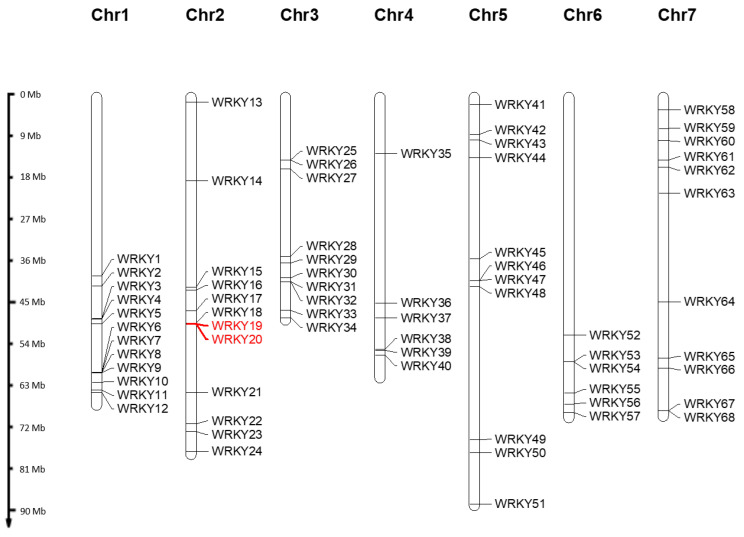
Positions of the *RcWRKYs* on chromosomes according to the *R. chinensis* genome of Hibrand Saint-Oyant et al. [22]. Two sequences located only on the genome of Raymond et al. [17] are highlighted in red.

**Figure 2 plants-13-01066-f002:**
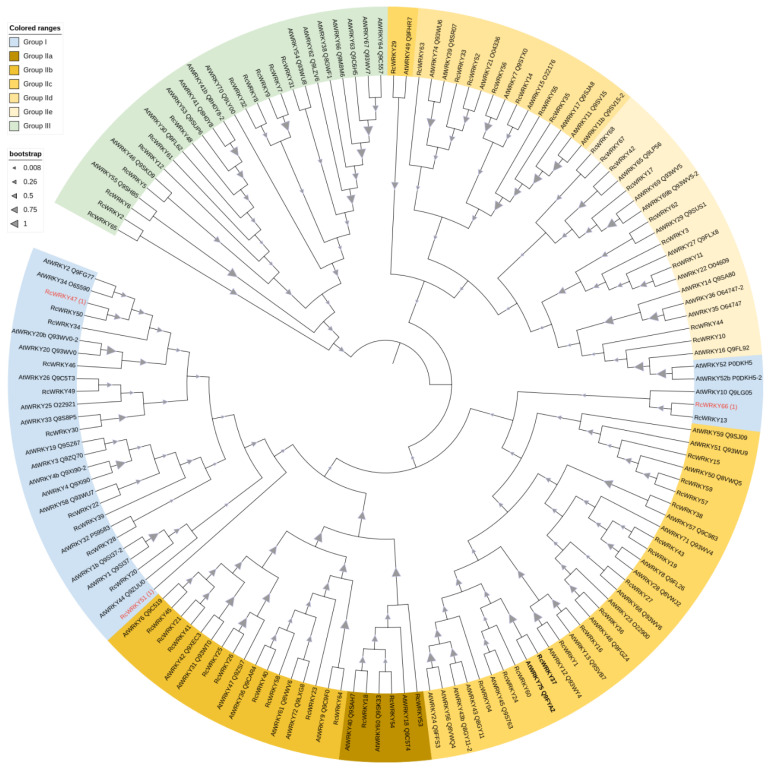
A phylogenetic tree of *A. thaliana* and *R. chinensis* WRKY genes. The tree was inferred using the maximum likelihood method and JTT matrix-based mode. The tree with the highest log likelihood (−138,959.49) is shown. The percentage of trees in which the associated sequences clustered together is illustrated by the triangles on the branches. This analysis revealed 149 WRKY amino acid sequences, of which 81 were from *Arabidopsis* and 68 were from *R. chinensis*. There were a total of 2992 positions in the final dataset. Evolutionary analyses were conducted in MEGA X, and the graphic was created with iTOL. Three *R. chinensis* sequences are marked in red because they were mapped to Group I but had only one WRKY domain.

**Figure 3 plants-13-01066-f003:**
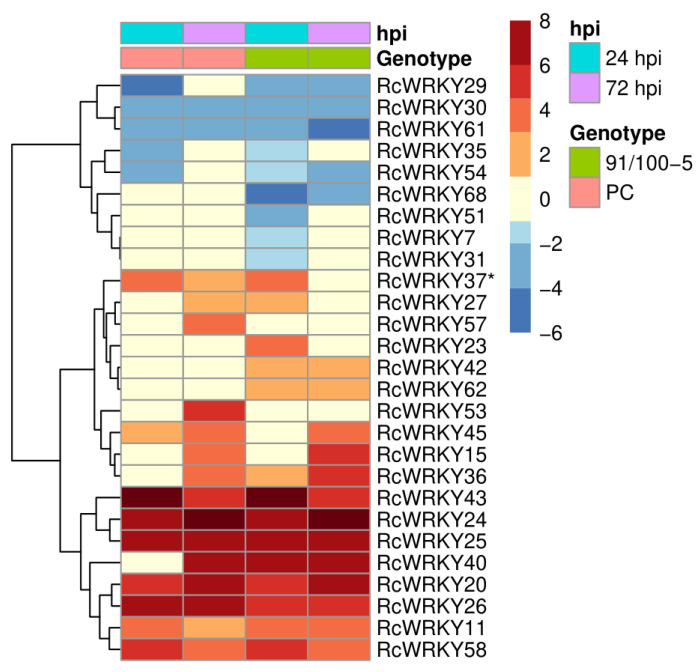
The transcriptional changes (log2-fold change) of 27 *WRKY* genes in rose leaves infected with *D. rosae* were compared to those in mock-inoculated leaves at 24 and 72 h post inoculation (hpi). The susceptible genotype ‘Pariser Charme’ (PC) and the *D. rosae*-resistant genotype 91/100-5 were used. Only the 27 *WRKY* genes that were differentially expressed between the inoculated leaves and the mock-inoculated leaves are shown and the gene *RcWRKY37*, described in more detail later, is marked with an * (*n* = 3).

**Figure 4 plants-13-01066-f004:**
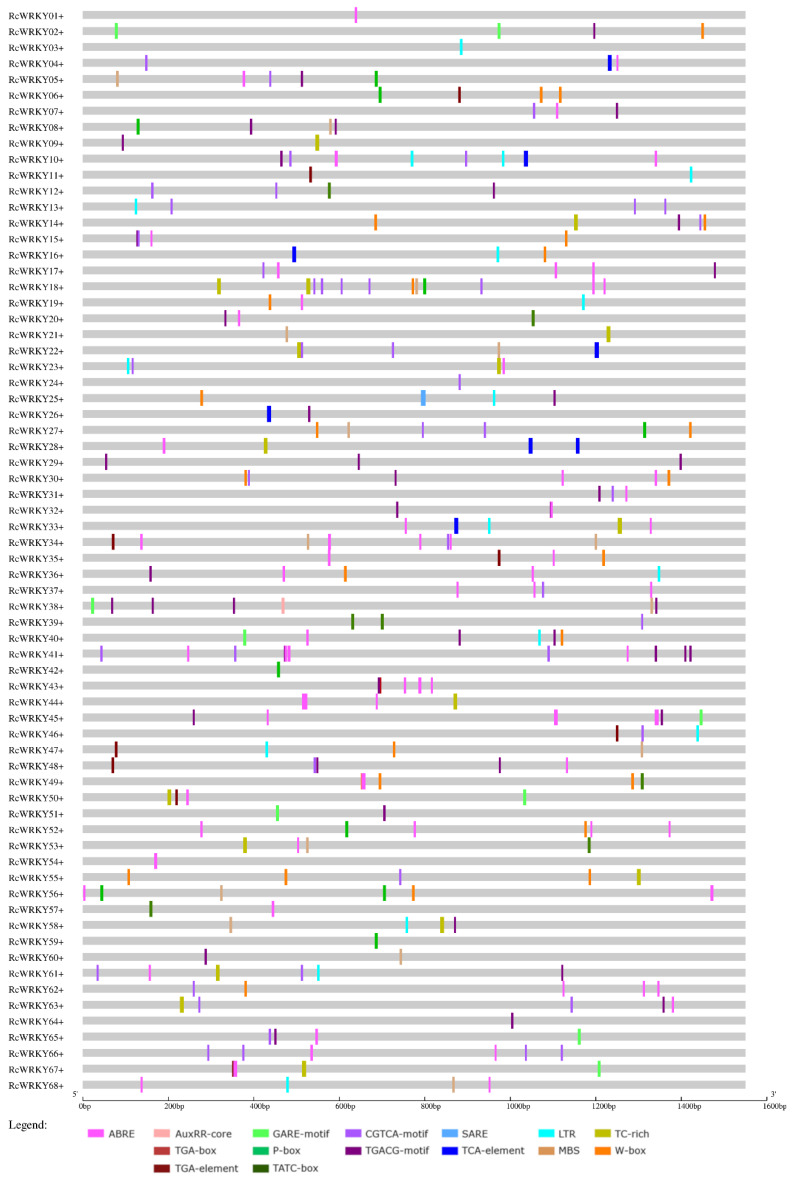
Promoter analysis of the region 1500 bp upstream of each gene on the plus strand.

**Figure 5 plants-13-01066-f005:**
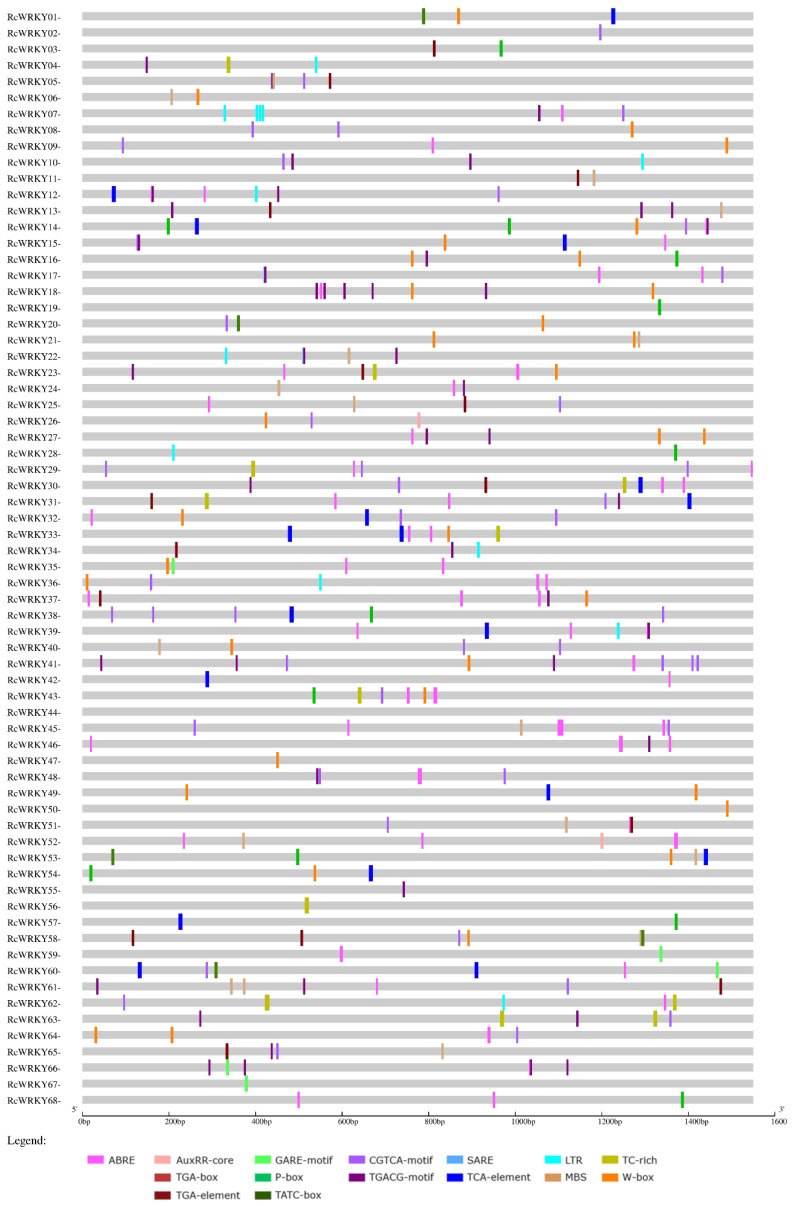
Promoter analysis of the region 1500 bp upstream of each gene on the minus strand.

**Figure 6 plants-13-01066-f006:**
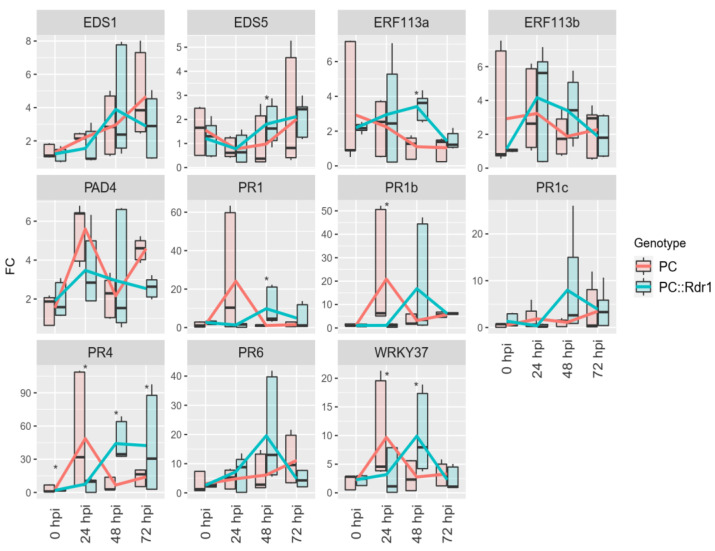
Changes in the expression of *RcWRKY37* and defence-related genes in ‘Pariser Charme’ (PC) and PC::*Rdr1* [31] leaves inoculated with *D. rosae*. Changes in expression were calculated for each time point relative to mock-inoculated leaves of the respective genotypes. PC is susceptible and PC::*Rdr1* is resistant to *D. rosae*. The following genes were analysed as additional defence-related genes: *EDS 1* & *5*, *ERF113*, *PAD4*, *PR1*, *PR4*, and *PR6*. The samples were taken at 0, 24, 48, and 72 h post inoculation (hpi). Stars indicate significant differences between the two genotypes at the indicated points in time. The lines represent the fold change mean values, and the boxplots indicate the distribution of biological replicates (*n* = 3).

**Figure 7 plants-13-01066-f007:**
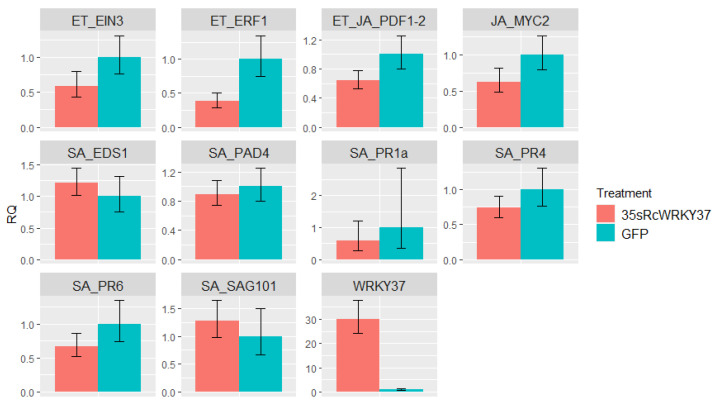
Gene expression of various defence-related genes after transient 35s:*WRKY37* overexpression relative to PC petals infiltrated with an empty vector (GFP). The data were obtained from three independent infiltration experiments with three biological replicates each. The gene names indicate their function in pathways related to ethylene (ET with *EIN3* and *ERF1*), jasmonic acid (JA with *PDF1-2* and *MYC2*), or salicylic acid (SA with *EDS1*, *PAD4*, *PR1*, *PR4*, *PR6*, and *SAG101*).

**Figure 8 plants-13-01066-f008:**
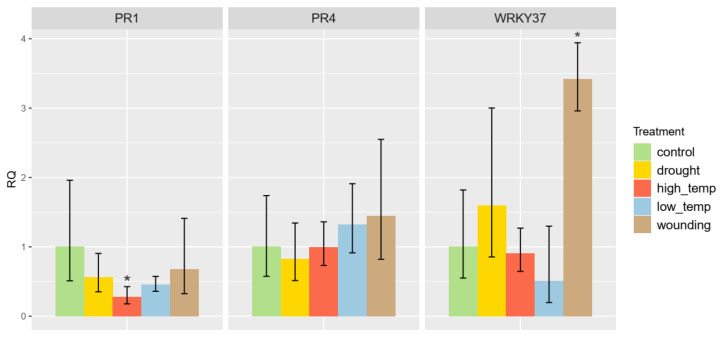
The expression levels of *RcPR1*, *RcPR4*, and *RcWRKY37* in PC leaves were measured in response to abiotic stress and compared to those in the control treatment. Significant changes in regulation with at least a 3-fold change are marked with an asterisk.

**Table 1 plants-13-01066-t001:** Promoter analysis results of differentially expressed WRKYs after *D. rosae* infection. The distribution of different motifs in up- and downregulated genes is summarised. The motifs are listed individually but also grouped according to their affiliation. The total number of up- or downregulated genes containing these motifs in their promoter sequences is also indicated.

Promoters of:		Upregulated *WRKYs* (Total Number: 18)			Downregulated *WRKYs* (Total Number: 9)			Nonregulated *WRKYs* (Total Number: 41)	
Motif in Connection with	Motif	No. of Motifs	No. of Sequences	Proportion of Sequences with These Motifs in the Promoter Region of Upregulated *WRKYs* (%).	No. of Motifs	No. of Sequences	Proportion of Sequences with These Motifs in the Promoter Region of Downregulated *WRKYs* (%).	No. of Sequences	Proportion of Sequences with These Motifs in the Promoter Region of Nonregulated *WRKYs* (%).
ABA	ABRE	43	15	83.3	23	9	100	30	73.2
Auxin (total)			7	38.9		5	55.6	15	36.6
	AuxRR-core	1	1	5.6	0	0	0	2	4.9
	TGA-box	1	1	5.6	0	0	0	2	4.9
	TGA-element	7	5	27.8	5	5	55.6	11	26.8
Gibberellin (total)			9	50.0		4	44.4	24	58.5
	GARE-motif	2	2	11.1	2	2	22.2	8	19.5
	P-box	6	6	33.3	2	2	22.2	12	29.3
	TATC-box	4	3	16.7	0	0	0	6	14.6
JA (total)			13	72.2		6	67	27	65.9
	CGTCA-motif	18	13	72.2	14	6	66.7	27	65.9
	TGACG-motif	16	12	66.7	13	6	66.7	27	65.9
SA (total)			6	33.3		3	33.3	15	36.6
	SARE	1	1	5.6	0	0	0	0	0
	TCA-element	5	5	27.8	3	3	33.3	14	34.1
Low temperature	LTR	9	8	44.4	7	3	33.3	13	31.7
Drought	MBS	11	9	50.0	4	3	33.3	14	34.1
Biotic stress (total)			13	72.2		6	66.7	27	65.9
	TC-rich	7	5	27.8	4	4	44.4	14	34.1
	W-box	19	13	72.2	5	3	33.3	20	48.8

## Data Availability

RNA-Seq raw data are available from NCBI under BioProject number PRJNA445241.

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
