# Peer review of "Genome-Wide Analysis of the WRKY Transcription Factor Family in Roses and Their Putative Role in Defence Signalling in the Rose–Blackspot Interaction"

_plants, 2024, doi:10.3390/plants13081066_

Round 1
Reviewer 1 Report
Comments and Suggestions for Authors
In this paper, the authors analyzed the composition of transcription factors of the WRKY family in roses. Through the analysis of expression level in response to the black spot fungus treatment, the authors screened candidate WRKY genes in the regulation of pathogen resistance; lastly, the authors carried out some expression analyses for RcWRKY37.
Despite the significance of research, I don't think this article can be accepted for publication as it current stands.
My main concerns are as follows: 1) the link between the experimental results and the conclusions of the article is too fragile, and too many of the results come from analogies with model plants, which cannot be the basis for the conclusions of the article. 2)The paper is written in a state of confusion that makes one wonder if the author was serious and rigorous when he wrote it.
I think the writer needs to first, carefully correct the ideas and presentation of the paper, from beginning to end. Otherwise, the thesis is very difficult to be understood.
Comments on the Quality of English LanguageSome of the paragraphs in the article are written in a somewhat hand-waving manner.
Author Response
Please find our response in the attached letter.

Reviewer 2 Report
Comments and Suggestions for Authors
Comments and Suggestions for Authors
Dear Author,
I have an honor to review the manuscript entitled “Genome-wide analysis of the WRKY transcription factor family in roses and their putative role in defense signalling in the rose-blackspot interaction” a research article submitted to MDPI Journal, Plants. Authors of this manuscript characterized 68 WRKY transcription factors (TFs) in rose. To characterize, they have performed a series of bioinformatic and molecular research and identified their roles in stress resistance and other physiological function in rose. Overall, the experiments are performed well and the results are convincing. Thus, the presented results takes up an important topic consistent with the profile of the Journal.
-However, even, manuscript is well organized and well described of the conception, I have some suggestions, which might improve the manuscript to make important to the wider audience.
-Few suggestions I have mentioned in the main text pdf file. Please check
-Major comments
-Firmed aim of the study that should be underlined precisely and simultaneously and highlight why this gene analysis is important to study in rose.
-There are many places where grammar can be improved. I suggest a careful revision by a professional language editing service. Extensive editing of English language and style is required.
Ref, formatting, organization do not match with Plants
Minor comments
-Title demonstrated; rose-blackspot interaction. However, there is no discussion about blackspot in the Introductory part. Improve
Abstract: -Good organization with results order.
L14: Abstract should not contain ref.
-Need scientific name of rose in abstract
L16-18; Very important line. But not giving specific clue and significant meaning. Please rewrite with exact findings
L19; what type regulation, not mentioned. Meaning is vague. Specify it
L29; different species of what?
Introduction:
-There is no introductory about WRKY37. Even, not for any other plants
-Introduction is not straightforward relating to results. Need substantial improvement.
2. Materials and Methods
-presentation is not continuous and smooth.
-Materials are not clearly described.
-L89; a line of this cultivar with the stable integrated resistance gene Rdr1 (PC::Rdr1, Menz et al. 2018) and the resistant rose genotype 91/100-5 (Debener et al. 2003) were used.-----------both are resistant. How they are differed? Resistant for what?
-Experiment should be reproducible. How inoculation was done? how samples were collected?
-what is hpi? Sample collected from where? Which part, which genotype?
-should maintain uniformity to write scientific name. also many where not italic
-2.4; first need to produce phylogenetic tree, then calculate. How it was developed?
-2.6; why suddenly only worky37 analyzed? Why not other? Need rationale
-L136,137; correct it
-for reproducibility, need to discuss PC infiltration procedure. Need agrobacterium strain name
L161; why darkness need?
-L165; what is under 2e?
3. Results
L171; who performed this genome annotation? You experiment or others? Mention it.
L185-191; meaning is not clear and conclusive
-you may mention exact no. of wrky found in each chr.
-L220-221; what is meaning grammar
3.5; you are working on 68 wrky genes. But checked only wrky37, why not others rather than many PR genes?
Fig. 6; what about mock under time point?
-need to mention 0-72hpi for all fig.
-you have results about PR genes, however, do not have any introductory. You may write some thing
- No where mentioned what are the defense related genes used? Not even fig. legend. It is very inconvenient explanation.
4. Discussion
L380; why one of the two. Mention specific one

Comments on the Quality of English LanguageModerate editing of English language required
Author Response

(The authors gave the same response as above.)

Reviewer 3 Report
Comments and Suggestions for Authors
Based 6 rose genomes (2 Rosa chinensis, R. multiflora, R. rugosa, R. roxburghii, R. sterilis), the authors extracted a set of putative WRKY genes, and analyzed the promoter regions and found numerous motifs related to induction by abiotic and biotic stress. From transcriptomic data of two rose genotypes inoculated with Diplocarpon rosae, authors found a total of 27 WRKY genes, and the RcWRKY37 was analyzed in detail. The results provided new evidence for the role of RcWRKY37 in rose signalling cascades, and provided new information for the study of WRKY genes.
1) Whether the WRKY genes found from the six genomes are all the same, whether there are duplications in different species, and how the 68 WRKY genes were finally determined, please describe them clearly.
2) It is recommended to supplement the photo results of Diplocarpon rosae inoculation.
3) The format of references is not consistent, please adjust it
Comments on the Quality of English LanguageQuality of English Language is good.
Author Response

(The authors gave the same response as above.)
